# Enhancing Healthcare Efficiency: The Relationship Between Effective Communication and Teamwork Among Nurses in Peru

**DOI:** 10.3390/nursrep15020059

**Published:** 2025-02-07

**Authors:** Monica Elisa Meneses-La-Riva, Víctor Hugo Fernández-Bedoya, Josefina Amanda Suyo-Vega, Hitler Giovanni Ocupa-Cabrera, Rosario Violeta Grijalva-Salazar, Giovanni di Deus Ocupa-Meneses

**Affiliations:** Grupo de Investigación “Innovación Humanizadora”, Universidad César Vallejo, Av. Alfredo Mendiola, Trujillo 13600, Peru; jsuyov1@ucv.edu.pe (J.A.S.-V.); hocupa@ucvvirtual.edu.pe (H.G.O.-C.); rgrijalvas@ucv.edu.pe (R.V.G.-S.); docupame@ucvvirtual.edu.pe (G.d.D.O.-M.)

**Keywords:** effective communication, teamwork, nurses, nursing, healthcare efficiency

## Abstract

Background: Effective communication in healthcare is essential for ensuring teamwork that is continuous, effective, and efficient. It plays a crucial role in supporting comprehensive and holistic care for patients while also guaranteeing the safety and satisfaction of the services provided. Objective: To determine the relationship between effective communication (as well as its dimensions: transmission of institutional culture, source of employee motivation, and facilitation of teamwork and conflict resolution) and teamwork among nurses in a national hospital in Peru. Methods: The research employed a quantitative, correlational, cross-sectional approach with a non-experimental design. The study population consisted of 328 nurses working in various hospital departments between January and October 2024. Data were collected using two questionnaires: The first assessed effective communication through three dimensions: transmission of institutional culture, source of employee motivation, and facilitation of teamwork and conflict resolution. The second instrument evaluated teamwork across three dimensions: institutional context, composition, and process. Results: A very strong positive correlation was observed between effective communication and teamwork among nurses, with a coefficient of 0.925 and a *p* value < 0.01. Conclusions: The study found a strong link between effective communication and teamwork among nurses. It underscores the role of communication, organizational culture, and motivation in strengthening teamwork, which enhances patient care and healthcare delivery. The findings highlight the impact of institutional culture, motivation, and conflict resolution, emphasizing soft skills and ethical behavior in improving team dynamics and organizational strategies.

## 1. Introduction

The World Health Organization (WHO) emphasizes the importance of improving the quality of healthcare services globally, highlighting the need for effective, integrated, and coordinated communication to address the growing demand for medical care [1]. Similarly, the United Nations (UN) promotes the third Sustainable Development Goal, which focuses on health and well-being. This goal aims to enhance the capacity of existing communication channels, strengthening credibility in social communication management and risk communication during health emergencies [2].

International objectives underscore the critical need for improving communication among healthcare professionals to prevent errors in verbal or telephone instructions. They advocate for the use of legal formats to ensure the correct understanding and documentation of messages, as communication failures can compromise patient safety [3]. Consequently, healthcare institutions in Latin America recognize effective communication as a strategic component of care, implementing mechanisms that encourage feedback within teams. Providing tools for clear and concise communication enhances outcomes and improves satisfaction in interactions. Effective communication creates a space where individuals interact and influence each other [4].

In this context, teamwork involves fostering a work environment where groups maintain strong performance in the services they provide [5]. International studies have highlighted the relationship between communication skills and nurses’ caregiving behaviors. Studies emphasize the importance of dialog and communication skills in the nursing process to ensure quality care [6]. Furthermore, communication plays a pivotal role in forming and ensuring the effective functioning of healthcare teams in critical areas [7]. Organizations strive to achieve operational and holistic outcomes that meet both internal and external needs. For healthcare organizations, delivering care at optimal quality levels has become a global priority. Achieving this requires implementing various components that work together to improve the quality of medical care. Collaborative teamwork, mutual trust, and commitment among team members are fundamental to reaching shared goals for the benefit of internal and external users [8].

Most nursing professionals align with essential criteria for internal communication during shift handovers, such as strategic vision, teamwork, responsibility, face-to-face communication, and confidentiality [9,10]. Additionally, individuals recognize progress in generic skills, particularly communication, and perceive the personal and academic benefits of these experiences [11]. On the other hand, nursing teams emphasize skills like empathy, commitment, and leadership as vital for team success. Despite challenges such as communication gaps, individualism, and temporary work arrangements, effective communication within teams reduces issues and fosters stronger bonds [12].

Interpersonal communication is often deficient, underscoring the need to strengthen this skill among nursing professionals. Competent execution of any professional role requires not only the acquisition and mastery of conceptual competencies but also the development of social skills that enable professionals to establish effective and rewarding relationships with their peers. In this regard, effective interpersonal communication is crucial for developing satisfactory skills in human relations within teamwork [13].

The effectiveness of communication is fundamental for promoting clear and precise interactions, facilitating problem-solving within work teams, and benefiting the institution. However, in the organization, challenges such as information dispersion due to rotating schedules and a lack of commitment to committees undermine cohesion and feedback among nursing staff [14].

Communication within nursing teams is essential for ensuring quality care but faces obstacles, such as workload overload and the lack of meetings, which can lead to errors. It is necessary to implement improvements in communication processes and human resources to foster a culture of organizational safety and strengthen collaboration and commitment among professionals [7,15]. It should be noted that Robbins and Judge’s theory identifies several limitations that hinder effective communication. These obstacles include filtering, selective perception, information overload, emotional influence, ambiguous language, maintaining silence, fear of communication, and dishonesty [16].

The concept of effective communication is defined as a process through which an environment conducive to interaction is created, allowing individuals to exchange ideas, feelings, and viewpoints [17]. Communication occurs when one person exerts influence over another, highlighting the importance of this process in human relationships [18].

There are three fundamental aspects of effective communication:(a)Transmission of Institutional Culture: This refers to the set of distinctive and stable traits that define the identity of each institution [15].(b)Source of Employee Motivation: Communication acts as a change agent that conveys organizational culture and adjusts according to the institution’s established policies [19]. Effective communication becomes a crucial element for motivating employees, creating an environment where they feel integrated [9].(c)Facilitation of Teamwork and Conflict Resolution: This is achieved through planned activities and clear communication of ideas [20]. This approach seeks to foster healthy interpersonal relationships among team members, using dialog as the primary tool for resolving any disagreements [21].

Teamwork is defined as the collaboration of a group of individuals who share a common goal and, by combining their efforts, can achieve superior results. Each member contributes their skills and competencies, resulting in a cumulative effort toward a more significant outcome [22]. It is crucial that each participant develops social skills to interact effectively with other team members [23]. However, in some groups, there is no significant collaboration among members, leading to a lack of positive synergy [20]. Teamwork establishes three aspects [17,24]:

Institutional Context: This is key to managing resources, distributing responsibilities, and encouraging staff participation to meet client demands. Organizational success depends on an environment that fosters efficient resource management and leadership that strengthens the team, promoting trust and positive outcomes [17].

Composition: Effective teams must include individuals with the necessary competencies to achieve optimal performance. It is recommended that these teams maintain a balance of individuals with diverse skills, similar to forming a team that seeks equilibrium among different roles to achieve high-quality outcomes [17,24].

Process: The importance of efficiency in work groups is emphasized, highlighting the need to set specific goals and foster interdependence, especially in large teams [25]. Work teams emerged to enhance organizational effectiveness and address challenges. An effective team promotes the achievement of shared objectives and evaluates the progress of joint results, which is essential for efficient performance in a specific area, especially in healthcare [26]. Interpersonal and organizational competencies at both individual and organizational levels facilitate the formation of cohesive teams capable of adapting and evolving to meet changing environmental demands [22,27,28].

In the context of nursing practice, various difficulties are encountered related to effective communication, teamwork, workload, and challenges in clear communication and collaboration due to a lack of available time [4,29]. Additionally, professional hierarchy in healthcare settings can hinder open communication, as nurses may feel reluctant to express their ideas or concerns to other team members, especially physicians or administrators. The lack of specific training in communication and teamwork skills during academic preparation can also affect collaboration and conflict resolution in the workplace. Resource shortages and insufficient institutional support, such as adequate technology and time to participate in teamwork activities, can also hinder effective communication and collaboration [30,31].

A culture that does not value open communication, collaboration, and teamwork can create barriers to effective practice. Furthermore, team diversity can pose challenges for communication and collaboration, but understanding and respecting individual differences are essential to overcoming these difficulties [32,33,34].

It is crucial to address this topic of communication and teamwork skills to foster a culture of innovation and promote organizational collaboration and open communication. Providing adequate resources and support can help overcome barriers to effective communication and successful teamwork in the nursing environment [35,36].

The objectives of the research were to determine the relationship between effective communication (as well as its dimensions: transmission of institutional culture, source of employee motivation, and facilitation of teamwork and conflict resolution) and teamwork among nurses in a national hospital in Peru.

As for the research hypotheses, they are detailed in the lines below, as well as in Figure 1.

General hypothesis: There is a relationship between effective communication and teamwork among nurses at a National Hospital in Peru.

Specific hypotheses:

There is a relationship between transmission of institutional culture and teamwork among nurses at a National Hospital in Peru.

There is a relationship between the source of employee motivation and teamwork among nurses at a National Hospital in Peru.

There is a relationship between the facilitation of teamwork and conflict resolution and teamwork among nurses at a National Hospital in Peru.

This study addresses the pressing issue of improving communication and teamwork among nurses in a national hospital in Peru, contributing to the broader healthcare literature by exploring how effective communication and teamwork impact the delivery of quality care. It uniquely examines the dimensions of communication, such as institutional culture, employee motivation, and conflict resolution, offering insights into how these factors can be optimized to enhance collaboration and patient outcomes in healthcare settings.

## 2. Methodology

The method used was a quantitative, correlational, cross-sectional approach with a non-experimental design [36]. A formal calculation of the required sample size was not performed prior to data collection. However, this study aimed to achieve a comprehensive representation of the nursing staff by including all eligible participants working at the national hospital. The inclusion criteria required participants to be currently employed as nurses at the designated hospital and to have a minimum of six months of continuous employment.

The total number of nurses employed at the hospital who met these inclusion criteria was 748. Among them, 328 nurses voluntarily participated in the study, representing 43.8% of the eligible population. This participation rate reflects a substantial proportion of the eligible nurses, contributing to the representativeness of the sample.

A total of 420 eligible nurses did not participate. The reasons for non-participation included lack of time, disinterest in the study, being unavailable during the data collection period due to vacation leave, medical leave, or work schedule conflicts, and not responding to the invitation to participate. This study was conducted between January and October 2024.

### 2.1. Instruments

The Effective Communication Scale [17] was used, which includes three dimensions:D1: Transmission of institutional cultureD2: Source of employee motivationD3: Facilitation of teamwork and conflict resolution

This scale consists of 25 items and employs a Likert-type scale. It was validated by experts, obtaining an Aiken’s V validity score of 0.9 and a reliability score of 0.8.

Additionally, the Teamwork Scale [17] was applied, which also includes three dimensions:D1: Institutional contextD2: CompositionD3: Process

This scale consists of 25 items and employs a Likert-type scale. It was validated by experts, achieving an Aiken’s V validity score of 0.9 and demonstrating reliability with a Cronbach’s alpha coefficient of 0.83, confirming its dependability.

These validated and reliable instruments were selected to ensure robust measurement of the variables under study. Their high validity and reliability scores confirm their suitability for evaluating effective communication and teamwork, which are central to understanding the dynamics within the organizational context examined in this research.

### 2.2. Procedures

After obtaining the necessary permissions from the institution’s authorities, coordination was conducted with the head of the nursing department to facilitate the execution of the project. This included providing staff lists and email addresses. Nurses were informed about the study and invited to participate. Upon obtaining their informed consent, a Google Forms link was shared for data collection, and communication was facilitated through Gmail and WhatsApp.

The collected data were organized into a matrix for review and subsequent analysis. Although measures were taken to exclude surveys that did not meet the established protocols, resulting in the formation of the final database, this was not necessary, as all the responses obtained were of good quality.

Throughout the process, ethical considerations were prioritized, ensuring the protection of authorship rights and the maintenance of scientific and professional integrity to uphold academic rigor. The authors received approval from the Ethics Committee, titled “Comité de Ética en Investigación de Maestría en Gestión de los Servicios de la Salud”, at Universidad César Vallejo. The approval was granted under the code 3387 on 4 December 2024.

### 2.3. Data Analysis

Data analysis was conducted using SPSS v.26. Descriptive and inferential analyses were performed, preceded by a Kolmogorov–Smirnov normality test, which revealed a non-normal distribution. Consequently, Spearman’s Rho correlation coefficient was used to investigate the relationship between the variables and their dimensions, and the results were presented in tables.

## 3. Results

The sociodemographic characteristics of the 328 nurses are detailed in Table 1. Most participants fall within the age groups of 31–40 years (45.00%) and 41–50 years (26.00%). A significant majority are women (82.00%), while 18.00% are men. Regarding marital status, 43.00% are single, and 38.00% are married. In terms of work experience, 34.00% have between 1 and 5 years of experience, and 26.00% have between 6 and 10 years.

Table 2 presents a descriptive analysis of the variables effective communication and teamwork.

(a)The effective communication variable has a mean of 99.34 and a standard deviation (SD) of 9.316, indicating moderate dispersion. Skewness is slightly positive (0.480), showing a small tendency towards lower values, while kurtosis is negative (−0.555), suggesting the distribution is somewhat flatter than a normal distribution.(b)The teamwork variable shows a mean of 99.02 and an SD of 8.959, also indicating moderate dispersion. Skewness is slightly positive (0.564), indicating a slight tendency towards lower values, and kurtosis is negative (−0.431), suggesting the distribution is slightly flatter than normal.

These high prevalence rates suggest strong perceptions of both communication and teamwork among the nurses.

Table 3 highlights the predominance of high levels for both variables.
(a)For effective communication, 70.73% of participants scored at a high level.(b)For teamwork, 75.61% scored at a high level.

Table 4 shows the relationships between the key variables studied:

**General Hypothesis**:There is a relationship between effective communication and teamwork among nurses at a National Hospital in Peru.

A very strong positive correlation was observed between effective communication and teamwork among nurses, with a coefficient of 0.925 and a *p* value < 0.01. This indicates a highly significant relationship, highlighting that effective communication is strongly linked to teamwork among nursing professionals.

**Specific Hypothesis 1**:There is a relationship between transmission of institutional culture and teamwork among nurses at a National Hospital in Peru.

Transmission of institutional culture and teamwork yielded a correlation of 0.795 with a *p* value < 0.01. This suggests that nurses who are effective in transmitting cultural values and norms are likely to foster stronger teamwork dynamics.

**Specific Hypothesis 2**:There is a relationship between the source of employee motivation and teamwork among nurses at a National Hospital in Peru.

Source of employee motivation showed a significant positive relationship with teamwork, with a correlation coefficient of 0.742 and a *p* value < 0.01. This indicates that both intrinsic and extrinsic motivation are key drivers of collaborative behavior among nursing teams.

**Specific Hypothesis 3**:There is a relationship between the facilitation of teamwork and conflict resolution and teamwork among nurses at a National Hospital in Peru.

Facilitation of teamwork and conflict resolution demonstrated a moderate positive relationship, the crucial role of effective communication within nursing teams, showing that it is not just about talking, it is about fostering connection, understanding, and collaboration. Nurses who can communicate well and share the institution’s values help create a strong sense of unity within the team, ensuring everyone is working towards the same goals. Motivation, whether internal or external, plays a big part too. Nurses who are motivated are more likely to actively engage with their colleagues, solve problems together, and contribute to a positive work environment. While resolving conflicts may not have the strongest correlation, it is still a key factor in helping teams navigate challenges and stay productive.

## 4. Discussion

The findings, based on a sample of 328 nurses, predominantly women aged 31–40, single, and with 1–5 years of work experience, highlighted that the variable effective communication was rated at high and medium levels in 70.73% and 29.27%, respectively. These results align with those of Hinostroza Veliz [30], who found that 80.3% considered effective communication to be good and 16.4% as average. Similarly, Calderón [17] reported that effective communication was moderate in 42.7%, deficient in 40.2%, and efficient in 17.1%.

Effective communication in nursing is of utmost importance for fostering open, horizontal, assertive, and trustful dialog to promote ethical and humanistic–scientific behaviors in the healthcare environment. Creating trust-based spaces characterized by collaboration, solidarity, and empathy, through adequate verbal and non-verbal communication, enables a focus on patient–family care. It is worth noting that communication is a fundamental element for enhancing or limiting the ability to address user problems and overcome adversities, particularly in the context of work overload in healthcare settings. The responsibility of nursing staff, primarily linked to administrative activities, often limits the time available for extended communication with patients and their families [31].

Regarding teamwork, high and medium levels prevailed at 75.61% and 24.39%, respectively. These results are consistent with Hernandez Chávez [32], who found that the appropriate level prevailed at 58%. Similarly, Calderón [17] reported that teamwork was moderate in 42.7%, deficient in 40.2%, and efficient in 17.1%. It can be inferred that teamwork in healthcare is not only effective in the recovery and treatment process of patients but also creates collaborative spaces that promote a positive and healthy work environment for health professionals, patients, and families. It is important to emphasize that the skills and competencies developed by healthcare professionals transcend into collaborative work, enabling them to resolve problems and overcome conflicts [32,33,34].

On the other hand, regarding the general objective, a highly positive correlation (r = 0.925) was found between effective communication and teamwork, with a *p* value less than 0.01. This evidence suggests that as communication improves, cohesion and collaboration within the nursing team are also strengthened. It is vital to highlight that healthcare professionals must develop soft skills, as well as ethical, scientific, and humanistic behaviors, to harmonize and enhance collaborative work among peers, fostering multidisciplinary teamwork to address adverse situations encountered in healthcare settings [33,34].

On the other hand, the dimension “Transmission of institutional culture” was found to have a significant relationship with teamwork, as evidenced by a coefficient of 0.795 (*p* < 0.01). These results differ from those of Calderón [17], who found a moderate positive correlation (r = 0.496). Consequently, these findings demonstrate a positive attitude towards assimilating and cooperating in knowledge about institutional protocols and norms, which need to be reinforced to achieve set goals and objectives. This indicates that nurses who effectively transmit the team’s cultural values and norms tend to foster a collaborative environment, which is essential for patient care. This underscores the importance of promoting an organizational culture that encourages shared values and a common vision, which is vital for cultivating a work environment that supports collaboration and mutual trust [35,36].

In the case of the “Source of employee motivation” dimension, a positive correlation of r = 0.742 (*p* < 0.01) was observed with teamwork. Calderón [17] reported a *p* value of 0.561, indicating a moderate positive correlation. This finding highlights that the intrinsic and extrinsic motivation of nursing professionals can be a determining factor in their ability to work collaboratively, which in turn impacts the quality of care provided. This suggests that motivation, both internal and external, is key to enhancing staff commitment and their ability to collaborate effectively. Healthcare institutions should focus on developing programs that promote staff motivation and well-being, which, in turn, positively influence team dynamics [4].

Finally, the dimension “Facilitation of teamwork and conflict resolution” showed a more moderate relationship, with a coefficient of 0.625 (*p* < 0.01). These results align with those of Calderón [17], who found a moderate positive correlation with a value of 0.628. Although this relationship is positive, its lower magnitude suggests that other factors may influence the team’s ability to solve problems effectively. These results are consistent with recent studies showing that communication and work culture are key elements for successful teamwork in healthcare settings. Implementing strategies to strengthen communication and staff motivation could be crucial to improving not only nursing team dynamics but also patient care outcomes. However, the lower magnitude of this correlation highlights the need for further investigation into factors that may influence team problem-solving. This suggests that, while communication and culture are essential, addressing other elements affecting team efficacy under high-pressure situations is equally critical [34].

## 5. Limitations

This study focused on a sample of 328 nurses working within a specific healthcare setting. While this group provided valuable insights, it may not fully capture the diversity of nursing professionals across different regions or institutions, potentially limiting the broader applicability of the findings.

The data were collected through self-reported questionnaires, which allowed participants to share their perspectives directly. However, this method can sometimes introduce challenges, such as a tendency to present responses in a favorable light (social desirability bias) or difficulties in accurately assessing personal behaviors and attitudes.

Although the study uncovered meaningful correlations between various dimensions and teamwork, it did not delve deeply into other external factors that might also shape these relationships. Elements like organizational policies, leadership styles, or workplace pressures could play a role and warrant further exploration.

Lastly, the results reflect the unique cultural and institutional context of the healthcare setting studied. Differences in healthcare systems, professional expectations, and cultural norms across other settings might influence how these findings apply elsewhere, highlighting the importance of considering local context when interpreting the results.

## 6. Contributions

This study provides valuable insights into the role of communication and teamwork in nursing within a national hospital in Lima, Peru. One of the study’s key strengths is the size and diversity of the sample, consisting of 328 nurses, which allows for the generalization of the findings within the institutional context. Additionally, the use of validated and reliable scales, such as the effective communication and teamwork scales, ensures that the data collected are accurate and consistent. The strong correlations found between effective communication and teamwork provide solid evidence supporting the positive relationship between these two variables, enriching research in the healthcare field.

Another important strength is that this study addresses various aspects of communication and teamwork, such as the transmission of institutional culture, employee motivation, and conflict resolution. By analyzing these factors separately, this study offers detailed insights into specific areas that can be targeted to enhance team dynamics and patient care outcomes. The ability to measure the strength of these relationships provides healthcare administrators with practical data to guide organizational strategies for improving communication and teamwork within nursing teams.

Moreover, the research highlights the importance of fostering an organizational culture that supports healthcare professionals, which is essential for building trust and collaboration among them. By recognizing the impact of both intrinsic and extrinsic motivation on teamwork, this study demonstrates the complexity of collaborative work and the need for comprehensive approaches to improve cooperation in healthcare settings.

The findings also emphasize the value of soft skills, ethical behaviors, and humanistic communication in healthcare, which deeply influence team effectiveness and patient outcomes. This approach aligns with current trends in healthcare that prioritize holistic care, underscoring the importance of empathy, compassion, and effective communication.

## 7. Conclusions

The findings demonstrate a correlation between communication and teamwork, supporting the notion that in healthcare settings, fluid communication is essential to strengthen interpersonal relationships among nurses and foster collaborative work. This, in turn, contributes to improving quality indicators and the therapeutic nurse-patient relationship.

Moreover, this study emphasizes that the interconnection between effective communication, organizational culture, and staff motivation within nursing teams enhances a healthy, safe, and trustworthy hospital environment. It is recommended that healthcare institutions implement strategies focused on improving these areas to optimize teamwork and, consequently, enhance care quality indicators, patient quality of life, and mental health.

## 8. Implications

These findings highlight the vital role of communication in fostering teamwork within nursing teams. When communication flows effectively, it strengthens relationships among nurses, enhances collaboration, and ultimately improves the quality of care provided to patients. Strong communication also helps build a more meaningful nurse–patient relationship, which is crucial for delivering compassionate and effective healthcare.

The results also emphasize the deep connection between effective communication, a positive organizational culture, and staff motivation. Together, these elements create a hospital environment that is healthy, safe, and supportive for both patients and healthcare professionals. By focusing on these areas, healthcare institutions can nurture a workplace where trust and teamwork thrive, directly benefiting the overall quality of care.

To achieve this, healthcare organizations are encouraged to implement strategies that strengthen these aspects. For example, offering communication training, fostering a shared sense of purpose, and creating motivational programs can have a significant impact. These efforts not only optimize teamwork but also enhance patient outcomes, improve quality of life, and support the mental well-being of healthcare staff.

This study also points to the need for continued research and education. Exploring how communication, culture, and motivation influence teamwork in different settings could provide even deeper insights. Additionally, nursing-education programs should focus on equipping future professionals with the communication and teamwork skills they need to excel in collaborative healthcare environments.

These findings remind us that human connection is at the heart of healthcare. Strengthening this connection through communication and collaboration can transform the way care is delivered, making it better for everyone involved.

To further enhance the effectiveness of communication and teamwork within nursing teams, healthcare organizations must adopt a multi-faceted approach. Implementing structured communication frameworks, such as daily briefings or debriefings, can ensure that all team members are aligned and informed. Additionally, encouraging interprofessional collaboration, where nurses, physicians, and other healthcare staff work together more cohesively, can foster an environment of shared responsibility and mutual respect. These practices not only improve teamwork but also promote a culture of continuous improvement, where communication is viewed as an ongoing process of learning and adaptation.

Moreover, healthcare managers and leaders play a crucial role in modeling and reinforcing effective communication behaviors. Leadership training programs should emphasize the importance of active listening, empathy, and clear articulation of expectations to create an environment where team members feel empowered to share ideas and concerns. Leaders must also recognize and reward collaborative behavior, reinforcing the connection between strong communication and successful teamwork.

Finally, future studies should aim to explore the long-term effects of communication and teamwork training on healthcare outcomes. While this study offers valuable insights into the immediate benefits of effective communication, a broader investigation into its sustained impact across different healthcare settings—such as emergency care, intensive care units, and outpatient services—would be beneficial. Additionally, research could focus on how technology can further enhance communication and teamwork in nursing, particularly in the context of remote consultations and electronic health records.

By taking these steps, healthcare institutions can create a robust system that supports collaboration and empowers healthcare professionals to deliver high-quality, compassionate care. A commitment to improving communication within nursing teams not only benefits healthcare staff but also leads to better patient experiences and outcomes, ensuring that healthcare systems are equipped to meet the challenges of the future.

## Figures and Tables

**Figure 1 nursrep-15-00059-f001:**
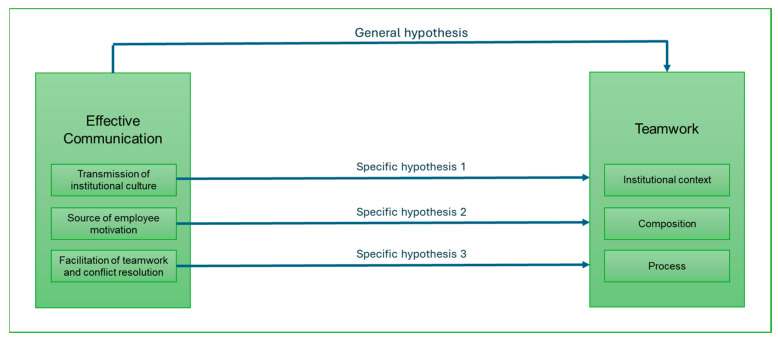
Theorical framework.

**Table 1 nursrep-15-00059-t001:** Percentage distribution of nurses by sociodemographic variables.

Variables	Scales	Frequency	% (n = 328)
Age	20 to 30	62	19%
	31 to 40	148	45%
	41 to50	86	26%
	51 to 60	23	7%
	≥61	9	3%
Gender	Female	268	82%
	Male	60	18%
Marital Status	Single	142	43%
	Partnered	58	18%
	Married	123	38%
	Widowed	5	1%
Years of Experience	1 to 5	112	34%
	6 to 10	84	26%
	11 to 15	72	22%
	16 to 20	31	10%
	>20	29	8%

**Table 2 nursrep-15-00059-t002:** Descriptive analysis of the variables: effective communication and teamwork.

Variables	Mean	Standard Deviation	Skewness	Kurtosis
Effective Communication	99.34	9.316	0.480	−0.555
Teamwork	99.02	8.959	0.564	−0.431

**Table 3 nursrep-15-00059-t003:** Levels of the variables effective communication and teamwork.

Variables	Level	Frequency	% (n = 328)
Effective communication	Medium	96	29.27%
	High	232	70.73%
Teamwork	Medium	80	24.39%
	High	248	75.61%

**Table 4 nursrep-15-00059-t004:** Correlation analysis between effective communication (and its dimensions) and teamwork.

Variables	Effective Communication	D1: Transmission of Institutional Culture	D2: Source of Employee Motivation	D3: Facilitation of Teamwork and Conflict Resolution
Teamwork	r = 0.925; *p* < 0.01	r = 0.795; *p* < 0.01	r = 0.742; *p* < 0.01	r = 0.625; *p* < 0.01

## Data Availability

Data will be provided by the authors if asked.

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
