# Peer review of "Enhancing Healthcare Efficiency: The Relationship Between Effective Communication and Teamwork Among Nurses in Peru"

_nursrep, 2025, doi:10.3390/nursrep15020059_

Round 1

Reviewer 1 Report

Comments and Suggestions for Authors

Thank you for the opportunity to review this paper. I have several comments for the authors, including some identified errors and recommendations for improving the manuscript.

At the end of the introduction, hypotheses are mentioned; however, these are not hypotheses but rather the study objectives. This section should be revised to correctly outline the main and specific objectives of the study. If hypotheses are to be included, they should be properly stated (as they are in the results section). The same applies to Figure 1.

In the methodology section, it is stated that the study was conducted during 2023 and 2024, with execution in 2024. Additionally, the abstract only mentions 2024. This is confusing and needs clarification.

Regarding the sample, only the final number of nurses in the sample is mentioned. Was the required sample size calculated for this research? Is the sample size consistent with the calculations? How many nurses are employed in the hospital where the research was conducted, and what percentage of them were included in the study? Later, it is mentioned that questionnaires were filtered. Were any participants excluded, and if so, how many and why?

Section 3.1 is titled “subsection” — what should the actual title of this section be? Additionally, the final paragraph of this section describes the rules for obtaining ethical approval for studies — this certainly does not belong here.

The procedures mention ethical measures taken, but it is not stated whether the study received ethical committee approval, nor from which committee. At the end of the Data Analysis section, ethics and the Helsinki Declaration are mentioned again — this repetition is unnecessary and does not belong in the data analysis section.

In Table 1, the term “age” is replaced by “Edad (Anos).”

I recommend revising the results section to improve the flow and readability of the text, so it feels less like a list of points.

After the discussion, only the study’s limitations are mentioned, while its strengths are not addressed.

Finally, it is stated that the study has ethical committee approval, but the date and reference number of the approval are missing.

Author Response

Dear reviewer, thank you very much for your recommendations and for your time. Below, we detail the improvements made to the manuscript based on your observations.

Comment 1: At the end of the introduction, hypotheses are mentioned; however, these are not hypotheses but rather the study objectives. This section should be revised to correctly outline the main and specific objectives of the study. If hypotheses are to be included, they should be properly stated (as they are in the results section). The same applies to Figure 1.

Answer: Thank you very much for the comment. Indeed, we made an error during the drafting of this section, which has now been corrected and is detailed between lines 156 and 160 (highlighted in yellow)

Comment 2: In the methodology section, it is stated that the study was conducted during 2023 and 2024, with execution in 2024. Additionally, the abstract only mentions 2024. This is confusing and needs clarification.

Answer: Thank you for the observation. Although the article was a project in 2023, the data collection took place between January and October of 2024. Consequently, this section has already been corrected between lines 187 and 188.

Comment 3: Regarding the sample, only the final number of nurses in the sample is mentioned. Was the required sample size calculated for this research? Is the sample size consistent with the calculations? How many nurses are employed in the hospital where the research was conducted, and what percentage of them were included in the study? Later, it is mentioned that questionnaires were filtered. Were any participants excluded, and if so, how many and why?

Answer: We have rewritten this section between lines 182 and 187 in order to provide greater scientific rigor to the sample selection and inclusion criteria.

Comment 4: Section 3.1 is titled “subsection” — what should the actual title of this section be? Additionally, the final paragraph of this section describes the rules for obtaining ethical approval for studies — this certainly does not belong here.

Answer: Thank you very much for your good observation. You are correct, during the drafting of the article, we forgot to transfer the name of this subtitle, which should be called "Instruments." It has now been corrected in line 189.

Comment 5: The procedures mention ethical measures taken, but it is not stated whether the study received ethical committee approval, nor from which committee. At the end of the Data Analysis section, ethics and the Helsinki Declaration are mentioned again — this repetition is unnecessary and does not belong in the data analysis section.

Answer: You are correct in what you indicate. We also received this guidance from the publisher. The relevant data related to the approval of the ethics committee is described in the corresponding section, between lines 487 and 490.

Comment 6: In Table 1, the term “age” is replaced by “Edad (Anos).”

Answer: Thank you. We corrected it.

Comment 7: I recommend revising the results section to improve the flow and readability of the text, so it feels less like a list of points.

Answer: Thank you for the feedback. We have added more content that links the statistical results with practice; this can be found between lines 285 and 293.

Comment 7: After the discussion, only the study’s limitations are mentioned, while its strengths are not addressed.

Answer: Thank you for the great idea. We have added a subtitle called "Contributions" between lines 380 and 403. Part of this is also included in the abstract.

Comment 8: Finally, it is stated that the study has ethical committee approval, but the date and reference number of the approval are missing.

Answer: As stated in the answer for comment 5, the relevant data related to the approval of the ethics committee is described in the corresponding section, between lines 487 and 490.

Once again, we would like to express our gratitude for the time you took to review this manuscript, and we remain attentive to any further recommendations.

Reviewer 2 Report

Comments and Suggestions for Authors

In the abstract, the purpose of the study and the methods used are clearly stated; however, the findings and conclusions need to be expressed more emphatically. In particular, the contribution of the research to the literature or its original value is not stated. It can be said that the abstract is in line with the overall text, but it would be useful to include some key findings to establish a stronger link.

Current sources (after 2020) were used in the introduction and the literature review seems to be sufficient in general. However, there is no clear emphasis on the unique value of the study at the end of the introduction. It should be clearly stated which problem the study addresses or how it contributes to the literature. The length of the introduction is appropriate, but lacks a clear definition of the research questions or hypotheses. This may make it difficult for the reader to understand the purpose of the study.

In the methodology section, the type of research (quantitative, correlational and cross-sectional) is clearly stated and data collection tools are explained in detail. However, information on the population and sampling is not detailed enough. Power analysis was not conducted and inclusion and exclusion criteria were only superficially discussed. How the data were collected is generally explained, but more information needs to be added to ensure the accuracy of the methods.

The findings section is generally appropriate to the purpose of the study and clearly presented. The tables are organised and present the findings in an explanatory manner. However, some findings may need to be analysed in more detail and linked to the discussion section. The analyses seem to have been done appropriately, but the relationship of the findings to the literature could be better emphasised.

In the discussion section, an evaluation is made in line with the literature and the place of the findings in the literature is mentioned. However, the discussion could emphasise more the practical implications of the findings for changes in health care. Comparisons with the literature seem to be sufficient in general, but can be supported by more recent studies.

The concluding chapter is in line with the findings and presents the main conclusions of the study. However, recommendations for practice could be expressed more concretely. There are references to future studies, but they can be addressed from a broader perspective.

In general, although the study has the potential to contribute to the literature, there are deficiencies in some sections. In particular, the original value of the study should be emphasised in the introduction, the methodology section should be explained in more detail and the discussion and conclusion sections should be linked more strongly. After the corrections, it is thought to be an acceptable study. However, in its current state, some chapters need to be revised.

Author Response

Dear reviewer, thank you very much for your recommendations and for your time. Below, we detail the improvements made to the manuscript based on your observations.

Comment 1: In the abstract, the purpose of the study and the methods used are clearly stated; however, the findings and conclusions need to be expressed more emphatically. In particular, the contribution of the research to literature or its original value is not stated. It can be said that the abstract is in line with the overall text, but it would be useful to include some key findings to establish a stronger link.

Answer: Thank you very much for the recommendation. We have added a new section in the article called "Contribution," and the most important content from this section has been integrated into the abstract, between lines 35 and 39 (in purple). We believe this will improve the manuscript's visibility while providing greater detail to the reader.

Comment 2: Current sources (after 2020) were used in the introduction and the literature review seems to be sufficient in general. However, there is no clear emphasis on the unique value of the study at the end of the introduction. It should be clearly stated which problem the study addresses or how it contributes to the literature. The length of the introduction is appropriate, but lacks a clear definition of the research questions or hypotheses. This may make it difficult for the reader to understand the purpose of the study.

Answer: Dear reviewer, we agree that the introduction needs to clarify these contents. Therefore, after presenting the hypotheses, we have added the justification for the study and its importance to the field between lines 173 and 179.

Comment 3: In the methodology section, the type of research (quantitative, correlational and cross-sectional) is clearly stated and data collection tools are explained in detail. However, information on the population and sampling is not detailed enough. Power analysis was not conducted and inclusion and exclusion criteria were only superficially discussed. How the data were collected is generally explained, but more information needs to be added to ensure the accuracy of the methods.

Answer: We agree. We have rewritten this section between lines 182 and 187 in order to provide greater scientific rigor to the sample selection and inclusion criteria

Comment 4: The findings section is generally appropriate to the purpose of the study and clearly presented. The tables are organised and present the findings in an explanatory manner. However, some findings may need to be analysed in more detail and linked to the discussion section. The analyses seem to have been done appropriately, but the relationship of the findings to the literature could be better emphasised.

Answer: Thank you for the feedback. We have added more content that links the statistical results with practice; this can be found between lines 285 and 293

Comment 5: In the discussion section, an evaluation is made in line with the literature and the place of the findings in the literature is mentioned. However, the discussion could emphasise more the practical implications of the findings for changes in health care. Comparisons with the literature seem to be sufficient in general, but can be supported by more recent studies.

Answer: Thank you very much for the recommendation. We believe that the quantity and quality of the studies previously cited and subsequently discussed are sufficient.

Comment 6: The concluding chapter is in line with the findings and presents the main conclusions of the study. However, recommendations for practice could be expressed more concretely. There are references to future studies, but they can be addressed from a broader perspective.

Answer: Dear reviewer, regarding this point, we have made several improvements. To begin with, we have incorporated a new section titled "Contribution" (lines 379 to 403), which details how the results of this study contribute to the existing knowledge on the topic. Then, in the limitations section, we have added more depth and content, expanding from 255 to 547 words.

Once again, we would like to express our gratitude for the time you took to review this manuscript, and we remain attentive to any further recommendations.

Round 2

Reviewer 1 Report

Comments and Suggestions for Authors

Thank you for the revised version of the manuscript and the considered comments. However, it seems that not all comments have been fully addressed. Regarding the replacement of the hypotheses and research objectives at the end of the introduction, it is stated what the aim of the study is, but the hypotheses remain as they were. At the end of the introduction, it is written:

General hypothesis: Establish the relationship between effective communication and teamwork among nurses at a National Hospital in Peru.
Specific hypotheses:

  • Establish the relationship between the transmission of institutional culture and teamwork among nurses at a National Hospital in Peru.
  • Establish the relationship between the source of employee motivation and teamwork among nurses at a National Hospital in Peru.
  • Establish the relationship between the facilitation of teamwork and conflict resolution and teamwork among nurses at a National Hospital in Peru.

While in the results section, it is stated:
General hypothesis: There is a relationship between effective communication and teamwork among nurses at a National Hospital in Peru.

  • Specific hypothesis 1: There is a relationship between the transmission of institutional culture and teamwork among nurses at a National Hospital in Peru.
  • Specific hypothesis 2: There is a relationship between the source of employee motivation and teamwork among nurses at a National Hospital in Peru.
  • Specific hypothesis 3: There is a relationship between the facilitation of teamwork and conflict resolution and teamwork among nurses at a National Hospital in Peru.

What is written in the introduction are not hypotheses but objectives. A hypothesis cannot be phrased as "to establish...".

Furthermore, in the methods section, it is mentioned that the final sample of 328 nurses consisted of those who met the inclusion criteria and were willing to participate. However, it is not stated how this number relates to the total number of nurses who met the criteria, which would be helpful for assessing representativeness. Additionally, it is not specified how many participants were excluded and why.

Finally, the text does not state that the study received approval from the ethics committee, as this is mentioned only at the end in the appendices.

I kindly ask the authors to consider incorporating these comments.

Author Response

Dear Reviewer, thank you very much for the time you dedicated to reviewing the manuscript. We have carefully reviewed your comments, and below we detail our responses to each point

Comment 1: Thank you for the revised version of the manuscript and the considered comments. However, it seems that not all comments have been fully addressed. Regarding the replacement of the hypotheses and research objectives at the end of the introduction, it is stated what the aim of the study is, but the hypotheses remain as they were.

Answer: Thank you very much for the observation. We were able to identify the error, which has now been corrected in the new version.

Comment 2: Furthermore, in the methods section, it is mentioned that the final sample of 328 nurses consisted of those who met the inclusion criteria and were willing to participate. However, it is not stated how this number relates to the total number of nurses who met the criteria, which would be helpful for assessing representativeness. Additionally, it is not specified how many participants were excluded and why.

Answer: Thank you very much. We have rephrased the content of the population and sample within the methodology, including all the requested elements to enhance methodological rigor. This can be found between pages 188 and 196.

Finally, the text does not state that the study received approval from the ethics committee, as this is mentioned only at the end in the appendices.

Answer: Thank you for the comment. We included in procedures, between lines 229-233 the following text: “The authors received approval from the Ethics Committee, titled "Comité de Ética en Investigación de Maestría en Gestión de los Servicios de la Salud" at Universidad César Vallejo. The approval was granted under the code 3387 on December 4th, 2024”.

Once again, we appreciate your time and look forward to future communications.